# Epidemiology of Sports-Related Injuries and Associated Risk Factors in Adolescent Athletes: An Injury Surveillance

**DOI:** 10.3390/ijerph18094857

**Published:** 2021-05-02

**Authors:** Pablo Prieto-González, Jose Luis Martínez-Castillo, Luis Miguel Fernández-Galván, Arturo Casado, Sergio Soporki, Jorge Sánchez-Infante

**Affiliations:** 1Health and Physical Education Department, Prince Sultan University, Riyadh 11586, Saudi Arabia; pprieto@psu.edu.sa; 2Nuestra Señora de las Mercedes School, Tarancón, 16400 Cuenca, Spain; luisalba78@hotmail.com; 3Education Faculty, Autonomous University of Madrid, 28049 Madrid, Spain; luisdepucela@gmail.com; 4Center for Sport studies, Rey Juan Carlos University, 28028 Madrid, Spain; arturo.casado@urjc.es; 5ITV Secondary School, Santa Rosa, La Pampa 6300, Argentina; sergiosoporki@gmail.com; 6Performance and Sport Rehabilitation Laboratory, Faculty of Sport Sciences, University of Castilla-La Mancha, 45071 Toledo, Spain

**Keywords:** sports injuries, epidemiology, injury rate, adolescent athletes

## Abstract

The present study aimed to determine the epidemiology of sport-related injuries in amateur and professional adolescent athletes and the incidence of different risk factors on those injuries. Four hundred ninety-eight athletes aged 14 to 21 voluntarily participated in this prospective injury surveillance, conducted from 1 January 2019 to 31 December 2019. The information collected included: personal data, sports aspects, characteristics of the injuries, and lifestyle. Forty point four percent of the participants suffered an injury in 2019 (39% of them in a previously injured area). The average injury rate was 2.64 per 1000 h. Soccer presented the highest rate (7.21). The most common injuries were: lumbar muscle strains (12.24%), ankle sprains (11.98%), and bone fractures (9.31%). Ankles (36.12%), knees (19.32%), and shoulders (6.47%) concentrated the highest number of injuries. Fifty-nine point twenty-eight percent of the injuries occurred during practices, and 40.72% during competition or peri-competition. Higher injury rates were associated (in this order) with the following factors: (a) Greater number of hours of practice per week. (b) Not performing warm-ups. (c) Using inadequate sports facilities. (d) Being aged 14–17. (e) Not performing physical preparation. (f) Inappropriate training load. (g) Not performing injury-preventive activities. (h) Performing sports technique without the supervision of one sports coach. (i) Inadequate sports equipment. In conclusion, since most injury risk factors are modifiable, it is imperative to implement strategies to reduce amateur and professional adolescent athletes’ injury rates.

## 1. Introduction

The practice of sport by adolescents generates physiological, psychological, and social benefits. These include improved health conditions, self-esteem, social interactions, and decreased risk of depression [1]. However, sports practice is inevitably linked with the appearance of injuries [2]. Moreover, this circumstance is aggravated by the increasing sports participation among adolescents in recent years [3].

Only in the United States, it was reported that 3.5 million youth under the age of 15 years old received medical care each year for injuries that occurred during sports practice. In addition, two-thirds of those injuries required care in emergency units [4]. LeBrun et al. [5] estimated that 23 million adolescents suffer sports injuries annually on the African continent. Similarly, Merkel [4] indicates that the estimated annual cost derived from sports injury management amounts to two billion dollars in the United States healthcare system. Furthermore, Knowles [6] mentions a prospective study conducted in the United States, wherein the annual statewide cost estimated of high school athletes was $9.9 million in medical expense, $44.7 million in human capital, and $144.6 million in total cost. In the case of Spain, no recent studies have analyzed the total cost of sports-related injuries in adolescents.

Similarly, an increase in the injury rates was observed in recent years in soccer [7]. This higher incidence is often attributed to a greater level of sports specialization and more intense practice at an early age [8]. This theory is supported by the fact that young athletes’ injury patterns seen in recent years are similar to those observed in mature athletes [8]. Under these circumstances, it is urgent to design strategies to reduce the incidence of sports injuries in adolescents due to their high cost in economic terms and the overload of health systems [7].

Likewise, reducing the risk of suffering injuries in such a significant population group will reduce youth sports attrition, promote lifetime participation in sports, and produce improvements in public health associated with the regular practice of sports [7]. In this regard, it is essential to highlight that there is a clear tendency to abandon sports practice during adolescence due to intrapersonal, interpersonal, and structural constraints [9].

Merkel [4] states that the sports drop-out rate of 15-year-olds is between 70% and 80%. Therefore, since adolescence is divided into three phases (early: 10–13 years; middle: 14–17; late: 18–21) [10], the highest drop-out rate occurs in middle adolescence. Moreover, according to the current evidence, one of the main reasons for sport drop-out is the occurrence of injuries [11]. Therefore, an in-depth review of the youth sports programs to make sports practice safer.

Importantly, promoting the design of strategies aimed at preventing injuries, protecting young athletes’ health, and increasing sports safety requires continuous surveillance of sports injury prevalence and patterns [12]. However, comparing the epidemiological results of existing research is complicated due to the different characteristics between studies [3,13]. Significant discrepancies in incidence rates among different research are common. Those discrepancies result from the differences in the target population, sports studied, country, competitive level, age group, and study type [3]. Thus, more epidemiological studies that consider all these variables are required to improve scientific knowledge about sports injuries and facilitate preventive interventions [14].

As for the current scientific research available related to sports injuries epidemiology and patterns in adolescents, the number of studies available is limited in the case of Spain. Moreover, most of them focused on specific sports (soccer, basketball, skateboarding, martial arts, padel), adult population, or recreational sports [15,16,17,18,19,20]. There is only one recent study on sports injuries in adolescents. However, it is a cross-sectional retrospective study focused on school sports [21]. 

In this context, it is necessary to know the sports-related injury epidemiology and patterns in adolescent athletes. It must be clarified which sports present the highest injury rates, the most frequent injuries, in what context the injuries occur, and if males suffer more injuries than females, and professional athletes more injuries than their amateur counter partners [14]. It is also important to determine the impact of different injury risk factors such as training load, sports technique, age, BMI, weekly hours of practice, sports equipment and facilities, performing injury-preventive activities, physical preparation, nutrition, and stress. Knowing this data is crucial for estimating the extent and cost of sports injuries. It will also increase the athletes’ safety and will be helpful to design more effective injury prevention strategies in the future. 

It is important to highlight that injury prevention programs reduce the incidence of injuries in adolescent athletes. However, these programs are multifaceted and imply the adoption of numerous preventive measures, the development of different fitness components, and the modification of sports technique or lifestyle habits. Therefore, it is unknown precisely which aspects of the preventive programs are crucial to prevent sports injuries [22]. Therefore, the objective of the present study was twofold: (i) to determine the epidemiology of sport-related injuries in amateur and professional athletes in the Autonomous Community of Castilla-La Mancha (Spain); and (ii) to analyze the incidence of different risk factors on those injuries.

## 2. Methodology 

The current research data collection process was organized based on the checklist items for reporting observational studies on injury and illness in sports designed by the International Olympic Committee Consensus Statement [23]. The following steps were taken.

### 2.1. Study Design

A prospective injury surveillance was conducted from 1 January 2019 to 31 December 2019.

### 2.2. Setting

The study was carried out in the Autonomous Community of Castile-La Mancha (Spain). It is an inland region located in the southern half of the Iberian Peninsula. In 2020, it had a population of 2,045,211 inhabitants [24]. 

### 2.3. Participants

Of a total of 1547 subjects invited to participate in the present study, 945 met the inclusion criteria, and finally, 498 completed it (flow chart for selection of study participants is shown in Figure 1). The inclusion criteria were as follows: (a) Being aged between 14 and 21 years old. (b) Being a professional or amateur athlete. (c) Practice one sport for at least five hours a week within a sports club, and have practiced that sport for at least two years. (d) Do not suffer any injury or disease incompatible with their sport’s practice at the beginning of the study.

Study participants received precise information about the objectives, benefits, and risks associated with their inclusion in the present study. All subjects signed an informed consent form declaring their willingness to be included in it. Additionally, the parents of the participants under 18 years signed an informed consent authorizing their children to participate in the study.

Athletes and their coaches and physical trainers were invited to participate in the study through the athletes´ Physical Education and Sports teachers. They were recruited from 10 Educational Institutions. Eight were Secondary Schools (four public and four private) located in rural and urban areas. The two other educational institutions were one public University (with campuses in six different cities of the Autonomous Community) and one private University.

### 2.4. Variables

Once the six leading researchers defined the study outcomes, an injury surveillance form was created to address all variables. The leading researchers listed and subsequently valued the items that could be included in the study from 0 to 10. Next, the Item-Level Content Validity Index (I-CVI) was calculated using the following formula:(I-CVI) = (Ne)/(Nt)(1)
where I-CVI: content Validity Index; Ne: number of experts giving a rating of “very relevant”; Nt: the total number of experts. Only the items with a score higher than 0.8 were included. As for reliability, the consistency of the results collected by the research assistants in quantitative variables was analyzed with Cronbach’s alpha. The value obtained was α = 0.84, which reflects a very good level of consistency.

Thereupon, the checklist was created. This document consisted of 24 items, grouped in four dimensions, as described in Table 1:

To facilitate the registration of non-categorical and non-quantitative variables (i.e., appropriate sports injury prevention programs, adequate warm-up, adequate sports equipment), a scale ranging from 1 to 5 was used. A value of 1 corresponded to absolutely appropriate/adequate, and 5 absolutely inappropriate/inadequate. Injury severity was also rated from 1 to 5, being 1 recovery without rehabilitation or medical advice, and 5 undergoing surgical intervention. Similarly, the stress level was rated from 1 to 5, with 1 being no stress and 5 being severe stress.

### 2.5. Data Sources and Measurement

Ten research assistants collected data. All of them were qualified Physical Education and Sports teachers working at the educational centers that voluntarily participated in the study.

### 2.6. Bias

Three online theoretical sessions were held to unify criteria to avoid the potential risk of biased and to guarantee the consistency of the information collected. Six researchers and 10 research assistants attended the meetings. Firstly, the concept of injury was clarified. To be included in the present study, an injury was defined as “any event that occurred during an organized competition or practice and requires attention from a health care provider” [25]. Similarly, a professional athlete was defined as “the subject who, by virtue of a relationship established on a regular basis, voluntarily practice sport within the scope of one organization, sports club or entity in return for remuneration” [26]. Then, the meaning of the 24 items that make up the injury surveillance was explained in detail, and precise instructions were given to the research assistants regarding the data collection. Once this process was completed, the researchers met with the research assistants to rectify the errors before sending the data for further statistical analysis if the information collected was flagged and incomplete. Research assistants were also required not to interfere with the athletes’ competition and practice to avoid altering their study involvement.

### 2.7. Study Size

The sample size was estimated with the following formula [27]:n = Z2p × qN/e2 (N − 1) + Z2p × q(2)
where n = sample size, N = population size, Z = confidence level, p = probability of success, q = probability of failure, e = confidence interval. 

The confidence level was set at 95%, the confidence interval at 5%, and the probability of success at 50%. After performing the calculation, it was determined that the minimum number of individuals required to have a representative sample of the studied population within the Autonomous Community of Castile-La Mancha was 364.

### 2.8. Ethical Clearance

The present study was conducted in accordance with the principles set out in the Helsinki Declaration. Moreover, it was also approved by the Institutional Review Board of the Bioethics Committee at Prince Sultan University in Riyadh, Saudi Arabia (approval no. PSU IRB-2018-010017).

## 3. Statistical Analysis

The normality of the data was assessed using the Kolmogorov–Smirnov test and homoscedasticity with the Levene’s tests. To establish comparations between two samples, Student’s *t*-test was used when the data followed a normal distribution. For more than two cohorts or conditions, one-way ANOVA with Tukey´s post hoc test was conducted. In cases where the homogeneity of variance was violated, comparisons between two data samples were performed using Mann–Whitney U, whereas comparisons between more than two cohorts or conditions were conducted using Kruskal Wallis H, applying the Dunn–Bonferroni post hoc test for pairwise comparisons. When the data followed a normal distribution, the η2 parameter was used to estimate the effect size (ES). In addition, when the data did not follow a normal distribution, after conducting the Mann–Whitney U test, the ES was calculated using the following formula: ES = Z/√N(3)
where Z: Z-score; N: number of observations. The ES was interpreted as follows: 0.2 small effect, 0.5 moderate effect, and 0.8 large effect. The association between dependent and independent variables was examined using the Pearson product–moment correlation coefficient, and the results were interpreted as follows: r = 0 null correlation; 0.01 ≤ r ≤ 0.09 very weak, 0.10 ≤ r ≤ 0.29 weak, 0.30 ≤ r ≤ 0.49 moderate, 0.50 ≤ r ≤ 0.69 strong, and r ≤ 0.70 very strong. The injury rate was reported as the number of injuries per 1000 athlete-exposures (practice and competition) [28]. The significance level was set at 0.05. Data are presented as mean ± SEM. Statistical analysis was performed using SPSS, version 22.0 (SPSS, Inc., Chicago, IL, USA).

## 4. Results

Sample characteristics and epidemiological aspects are shown in Table 2. The 10 most practiced sports by the study participants were respectively: soccer, swimming, weight training, athletics, basketball, tennis, judo, paddle tennis, volleyball, and cycling. Of these 10 sports, soccer, judo, and basketball presented respectively the highest injury rate, and weight training, cycling, and swimming the lowest. The total number of injuries was 529. Seventy-five-point ninety-eight percent of them occurred to participants of three sports modalities: 68.81% footballers, 10.96% basketball players, and 7.75% Judokas. The average injury rate was 2.64 per 1000 h. The percentage of injured subjects in 2019 was 40.4% (38.8% were amateur and 46.4 professional). Thirty-nine percent of the subjects suffered an injury in a previously injured area. The injury recurrence was 5.11% higher in professional athletes. The most common injuries were, in this order, lumbar muscle strains, ankle sprains, and bone fractures. The most common body regions were, respectively, ankles, knees, and shoulders. More than two-thirds of the injuries occurred in the lower limbs. Fifty-nine point two eight percent of the injuries occurred during athletic training, and 40.72% while competing or performing peri-competition activities. The most common injury mechanisms were, in this order: (a) No identifiable single event (repetitive transfer of energy, overuse). (b) Acute non-contact trauma. (c) Direct contact with an object.

Comparisons between different conditions and subgroups within the sample are shown in Table 3. In this regard, it was observed that injury severity ((IS), injury severity score (ISS), and injury rate (IR) was significantly higher in the 14-to-17-year-old cohort than in the 18- to 21-year-old age group [(ID: *p* < 0.001; ES = 1.45); (ISS: *p* < 0.001; ES = 0.72); (IR: *p* < 0.001; ES= 0.76)]. The Kruskal–Wallis H revealed a significant main effect of BMI (*p* < 0.001). In addition, the Dunn–Bonferroni post hoc showed that normal-weight individuals presented an ID and an ISS significantly lower than overweight subjects [(ID: *p* < 0.001; ES = 1.14); (ISS: *p* = 0.029; ES = 1.16). No significant differences were found in ID and ISS between the underweight and normal-weight individuals and underweight and overweight subjects. Furthermore, no significant differences were found in IR between BMI categories.

The ID, ISS and IR of the individuals who practiced sports less than 10 h a week were significantly lower than those who practiced sports for more than 10 h [(ID: *p* < 0.001; ES = 1.13); (ISS: *p* < 0.001; ES = 0.84); (IR: *p* < 0.001; ES = 1.03)]. The subjects who perform an individualized and sport-specific physical preparation presented also a significantly lower ID, ISS and IR than those who did not [(ID: *p* < 0.001; ES = 1.59); (ISS: *p* < 0.017; ES = 0.49); (IR: *p* < 0.001 Es =1.58)]. Similarly, the athletes who perform specific activities aiming to prevent injuries presented a significantly lower ID (*p* < 0.001; ES = 1.12), ISS (*p* < 0.001; ES = 1.14), and IR (*p* < 0.001; ES = 1.10). Furthermore, the individuals who perform warm-ups adapted to their abilities and specific characteristics of the session had a significantly lower ID (*p* < 0.001; ES = 1.14), ISS (*p* < 0.001; ES = 1.18) and IR (*p* = 0.019; ES = 0.103).

The subjects who used sports equipment in good condition that was frequently inspected presented a significantly lower ID (*p* < 0.001; ES = 2.71), ISS (*p* = 0.020; ES = 0.685), and IR (*p* < 0.001; ES= 2.32) than those subjects who did not. In the same vein, a significantly lower ID, ISS and IR was observed among the athletes who used sports facilities in good condition that were free of obstructions, well-lit, ventilated, and possessed adequate flooring compare with those who did not [(ID: *p* < 0.001; ES = 2.32); (ISS: *p* < 0.001; ES = 2.37); (IR: *p* < 0.001; ES = 2.27)]. The individuals whose training load was adapted to their abilities also presented a significantly lower ID (*p* < 0.001; ES = 2.01), ISS (*p* < 0.001; ES = 0.27), and IR (*p* < 0.001; ES = 2.01) than those who did not. Finally, the athletes who performed the sports techniques under the supervision of a sports coach had a significantly lower ID (*p* < 0.001; ES = 1.61), ISS (*p* < 0.001; ES = 1.56), and IR (*p* < 0.001; ES = 3.077).

On the contrary, no significant differences were found in ID, ISS, and IR between subgroups for the following factors: (a) Sex (male and female). (b) Competitive level (professional and amateur). (c) Following a healthy, balanced, and individualized diet. (d) Being subjected to stress due to the pressure received from family members, coaches, friends, or self-imposed.

The association between independent and dependent variables is shown in Table 4. As for the ID, a significant negative correlation was observed between this variable and the following aspects: (a) Performing specific activities aiming to prevent injuries. (b) Training load adapted to athlete’s ability. (c) The correct execution of sports techniques was supervised by one sports coach. The *r* values indicate that in all cases, the correlation was weak.

The ISS presented a significant negative correlation with the following factors: (a) Carrying out an individualized and sport-specific physical preparation. (b) Performing specific activities aiming to prevent injuries. (c) Performing warm-ups adapted to athletes’ abilities and specific characteristics of the session. (d) Using sports equipment in good condition and frequently inspected. (e) The correct execution of sports techniques was supervised by one sports coach. Moreover, a significant positive correlation was found between ISS and weekly hours of practice. Nevertheless, in all cases, the correlation between the ISS and the injury risk factors was weak.

The IR presented a negative correlation with the following variables: (a) Age. (b) Carrying out an individualized and sport-specific physical preparation. (c) Performing specific activities aiming to prevent injuries. (d) Performing warm-ups adapted to athletes’ abilities and specific characteristics of the session. (e) Using sports equipment in good condition and frequently inspected. (f) The correct execution of sports techniques was supervised by a sports coach. Finally, a significant positive correlation was observed between IR and the following two variables: (a) Weekly hours of practice. (b) Being subjected to stress. The correlation between the IR and the injury risk factors was also weak. 

## 5. Discussion 

### 5.1. Epidemiological Aspects

One of the present study’s main findings was to verify that 40.4% of the subjects suffered an injury in 2019 (46.4% were professional and 38.8% amateur). The percentage of subjects injured in the present study was lower than the percentage observed by Danes-Daetz et al. [3] in Chilean University athletes in only six months (48.8%). It was also lower than the percentage registered by Pujals et al. [19] in Spanish athletes aged between 21 and 38 during one sport season (78.5%), and lower than the percentage observed by Martínez-de-Quel-Pérez et al. [21], who reported a percentage of 68.7% in secondary-school males, and 47.8% in secondary-school females during one academic year. Therefore, despite the number of injuries is lower than in previous research, we consider that the number of injuries found in the present study is still very high. Hence, it might be reduced in the future with injury prevention programs.

The percentage of subjects who suffered an injury in a previously injured area in the present study was 39% (37.4 in amateur athletes and 43.5% in professional athletes). This finding reveals the importance of complete healing after suffering sports injuries. However, the data can hardly be compared with the rates observed in similar scientific investigations due to the diversity of research protocols used. However, some studies have ascertained that a previous injury is a risk factor for subsequent injury [29,30].

The average injury rate (per 1000 playing and training hours) found in the present study was 2.64 per 1000 h. This figure is considerably lower than the rate observed by Arthur-Banning et al. [31] in practices in NCAA and club sports among college athletes, which was respectively 3.9 and 3.8. However, in the same study, they reported an injury rate during game competitions of 18.3 in club sports, 13.79 in NCAA, and 10.28 in intramural sports. Similarly, the injury rate found in this study was lower than the average injury rate (training and competition) found by Pujals et al. [19], which was 4.1.

On analyzing the injury rate by sport, it was observed that soccer presented the highest rate (7.21), followed by judo (4.82) and basketball (4.31). This suggests that contact sports involve an increased risk of injury. As for soccer, Watson [7] stated that injury rates in youth soccer range between 2.0 and 19.4. Therefore, the present study’s injury rate was relatively low since it was close to the lower limit of the incidence found in previous studies. We understand that the high IR of soccer is due to the following factors: (a) It is a contact sport. (b) It is played outdoors. This exposes soccer players to extreme weather conditions, such as cold in winter, heat in summer, and heavy rain in spring. (c) The characteristics of the soccer field and the use of soccer boots with studs can limit or impair lower limb movements, including rotation. (d) The ball is played mainly with the foot. In addition, very often, tackles are committed on the supporting leg. The injury rate found among judokas in this study is not comparable to the epidemiology data obtained in other studies with judokas since they focus exclusively on competition periods [32]. Regarding basketball, the current research’s injury rate was lower than the rate reported by Cumps et al. [33], which amounted to 6.0. The discrepancy between both figures could be due to the study mentioned above was conducted 12 years before the present study, and it is conceivable to think that the strategies to prevent injuries have improved. Furthermore, unlike in the present study, all participants were senior basketball players, which implies a higher volume and hours of practice, and a higher injury risk.

The most common injuries were: Lumbar muscle strains (65 injuries (12.28% of the total)), ankle sprains (63 injuries (11.91% of the total)), and bone fractures (49 injuries (9.31% of the total)). These results are consistent with Hootman et al. [34] and Trompeter et al. [35], since they also verify the high incidence of lumbar injuries and ankle sprains. By anatomical region, the three areas that concentrated the highest number of injuries were, respectively, ankles (36.12%), knees (19.32%), and shoulders (6.47%). These results agree with those obtained by Danes-Daetz et al. [3], who reported that the two most injured anatomical areas were ankles (24.1%) and knees (14.8%). The mentioned authors also observed that 63.0% of the injuries occurred in the lower extremities, 31.5% in the upper extremities, and 5.5% in the trunk. These results are similar to those obtained in the present study, in which 68.33% of the injuries occurred in the lower body, 20.45% in the upper limb, and 11.22% in the trunk. Habelt et al. [11] also observed a similar incidence (lower extremities 68.71%, upper extremities 25.27%, spine 2.57%, and head 1.99%).

Fifty-nine point twenty-eight percent of the injuries occurred during practices, and 40.72% during competition or peri-competition, reflecting that competition involves a higher injury risk. This matter has also been observed in previous studies [8]. The most common injury mechanisms observed in this study were no identifiable single events (repetitive transfer of energy, overuse) (30.64% of the cases) and acute non-contact trauma (21.44% of the cases). These results reflect that overuse injuries represent a high percentage of the total number of injuries reported. This finding is consistent with the study conducted by Patel et al. [8]. They state that overuse injury is often variable with the sport, but this type of injury is on the increase in adolescents due to the increased intensity in youth sports. 

### 5.2. Sex

A higher injury rate in males than females has frequently been found in the scientific literature [36,37]. Traditionally, the highest injury rate among males has been attributed to the socialization processes. It is assumed that males take more risk and are less protected than females [37]. However, in the present study, no significant differences were observed between sexes in terms of ID, ISS, and IR. Pujals et al. [19] also found no sex differences in athletes of 25 sport modalities. Therefore, the impact of the sex factor remains unclear and might be studied in future research.

### 5.3. Age

A significantly higher ID, ISS, and IR were observed in youth between 14 and 17 than those between 18 and 21. This result coincides with Patel et al. [8], who observed a higher IR among youth between 15 and 18. Sreekaarini et al. [38], in one study made with athletes, also observed a higher number of injuries in the age 14 years, followed by age 15 years. We consider that adolescents aged between 14 and 17 suffer more injuries due to the replication of adult sports training models in youth. In this respect, Hernán-Guzmán [39] attributes the increase in acute and subacute injuries in this age group to the increasing practice time and intensity in youth sports.

### 5.4. BMI

Although no significant correlation was found between ID, ISS and IR, and BMI, overweight study participants presented an ID and ISS significantly higher than normal-weight individuals. This finding agrees with the study of Richmond et al. [40]. They observed a greater injury risk in obese than in healthy-weight adolescents. Richmond et al. [41] also verified an increased risk of suffering a sports injury in obese adolescents. They attributed the increased injury risk to the greater absorption of forces by soft tissues and joints.

### 5.5. Competitive Level

No significant differences were found in ID, ISS, and IR between amateur and professional athletes in this research. This finding coincides with the results obtained by van Beijsterveldt et al. [42] in one study conducted with soccer players, where no significant differences were observed between amateur and professional individuals in the total number of injuries. Zurita Ortega et al. [43] also observed no significant differences in injury severity between professional and amateur athletes. However, they also reported that semi-professional athletes suffer fewer severe injuries. 

All these results are surprising since it could be expected that amateur athletes suffer fewer injuries. Nevertheless, Bahr and Krosshaug [22] understand that the high incidence of sports injuries found in amateur athletes corresponds to an increase in youth sports competitiveness. Another possible explanation would be that the injury prevention strategies used by professional athletes are better than those applied to amateur athletes.

### 5.6. Weekly Hours of Practice

The present study verified that the number of weekly hours of practice is a decisive factor, since athletes who practiced sports for less than 10 h had a significantly lower ID, ISS, and IR than those who practiced sports for 10 h or more. Furthermore, there was also a significant positive correlation between weekly hours of practice and ISS and IR, further reinforcing this fact. These results seem to be logical. The greater hours of practice, the greater the exposure, and the greater cumulative fatigue. In this respect, Johnston et al. [29] observed that an increased injury risk in endurance sports was associated with high training distances, training frequency, and low weekly and high monthly training durations. However, they also state a lack of research analyzing the effect of training volume on injury rate.

### 5.7. Performing an Individualized and Sport-Specific Physical Preparation

As could be expected, this factor has proved to be relevant in the present study to prevent sports injuries. Significant differences were observed between the subjects who carried out an individualized and sport-specific physical preparation and those who did not in ID, ISS, and IR. Likewise, performing an individualized and sport-specific physical preparation presented a significant negative correlation with ISS and IR. However, although one of the main objectives of physical conditioning is to prevent injuries, to our knowledge, the effect of this factor in injury prevention has not been examined in previous studies. Therefore, it should be evaluated in future research.

### 5.8. Performing Specific Activities Aiming to Prevent Injuries

This factor has proved to be crucial in the present study to prevent sports injuries. The athletes who performed specific activities aiming to prevent injuries presented a significantly lower ID, ISS, and IR than those who did not. Furthermore, performing specific activities to prevent injuries significantly negatively correlated with ID, ISS, and IR. These results are consistent with the study conducted by Hanlon et al. [44]. They verified that injury prevention programs reduce modifiable intrinsic risk factors in the lower extremities among young athletes. Nevertheless, they also caution that several intrinsic risk factors were not significantly affected or addressed in the mentioned injury prevention programs.

### 5.9. Performing Warm-Ups Adapted to Athletes’ Abilities and Specific Characteristics of the Upcoming Activity

This factor was also essential to prevent sports injuries, as could have been expected. Subjects who performed appropriate warm-up activities presented significantly lower ID, ISS, and IR than those who did not. Likewise, performing adequate warm-ups had a significant negative correlation with ID, ISS, and IR. This result coincides with similar studies carried out with adolescents [45,46,47]. Therefore, based on the current evidence, and considering that one of the warm-up’s main objectives is to prevent sports injuries, it seems clear that warm-up is a crucial aspect in reducing injury risk. Therefore, adequate warm-ups might always be performed before training sessions.

### 5.10. Sports Equipment in Good Condition and Frequently Inspected

The subjects who used adequate sports equipment presented a significantly lower ID, ISS, and IR. Besides, a significant negative correlation between this factor and ISS was observed. In agreement with this result, Patel et al. [8] and Hootman et al. [34] consider that adequate sports equipment effectively minimizes sports injuries. Similarly, Sreekaarini et al. [38] verified that 28.8% of the sports injuries reported in one study made with adolescents were related to the equipment used.

### 5.11. Sports Facilities in Good Condition, Free of Obstructions, Well-Lit, Ventilated, and with Adequate Flooring

Although no significant correlation was found between this factor and ID, ISS, and IR, the subjects who used adequate sports facilities presented a significantly lower ID, ISS, and IR than those who did not. Therefore, using appropriate sports facilities can also be essential to prevent sports injuries. Accordingly, Sharma and Parveen [48] state that sports facilities have a significant role in the prevention of sports injuries, and Sreekaarini et al. [38] reported that (in the study mentioned in the previous section), 47.6% of the sports injuries were due to the sports surface.

### 5.12. Training Load

Predictably, this factor has proven to be relevant to prevent sports injuries. Subjects who used a significantly lower training load adapted to athlete’s ability presented a significantly lower ID, ISS, and IR than those who did not. Moreover, using an adapted training load was negatively correlated with ID and IR. This result coincides with numerous studies emphasizing the need to monitor and adapt the training load in youth sports to prevent sports injuries [30].

### 5.13. Sports Technique

As expected, this factor proved to be crucial in preventing sports injuries since it presented a significant negative correlation with ID, ISS, and IR. Additionally, the study participants whose sports techniques one sports coach supervised execution had a significantly lower ID, ISS, and IR than those who did not. These results are also consistent with numerous studies indicating that correct technical execution is a crucial factor in sports injury prevention [11].

### 5.14. Following a Healthy, Balanced, and Individualized Diet

The present study verified that this factor did not influence ID, ISS, and IR. Actually, the role of nutrition in injury prevention is not entirely clear. Close et al. [49] point out that nutritional interventions could reduce the onset of acute injuries in track and field athletes. Zurita-Ortega et al. [41] add that prevention strategies must include nutritional interventions due to injuries’ multifactorial nature. However, randomized controlled studies are needed to verify the role of this factor in injury prevention.

### 5.15. Being Subjected to Stress Due to the Pressure Received from Family Members, Coaches, Friends, or Self-Imposed

The stress factor did not exert an evident influence on the occurrence of sports injuries. Stress was only positively correlated with IR. Furthermore, there were no significant differences in ID, ISS, and IR between the athletes subjected to stress and those who were not. In this respect, Sreekaarini et al. [38] verified that stress is a determining injury risk factor in adolescent athletes. Nippert and Smith [50] also indicated that psychological interventions could potentially reduce the occurrence of injuries. However, we consider more studies are needed to determine the real impact of this factor on ID, ISS, and IR.

### 5.16. Study Limitations and Future Research Lines

The main limitation of the present study was the difficulty of comparing its findings with other sports-related injury research. This is due to the important differences between studies in key aspects such as the definition of injury, methodological criteria used, target population, data reporting, variety of sports analyzed, and variety of sports practice contexts. Another limitation is that the study was conducted in one of the 17 regions comprising the country. Therefore, the results are not generalizable to the remaining 16 regions.

Future research should continue registering sports injury epidemiological data using standardized methods and measurements, such as the injury rate. The studies might be prospective and randomized, and the recommendations established by the International Olympic Committee Consensus Statement might be followed [21]. In this way, valid comparisons between studies could be made. Future studies should also examine the effect of certain injury risk factors, particularly those barely studied (training volume and sport-specific physical preparation), and on those whose influence remains unclear (nutrition, stress, and sex).

## 6. Conclusions

Forty point four percent of the participants suffered an injury in 2019. Although this figure was lower than in previous recent studies, it is still very high considering the economic and human burden of sports injuries. Thirty-nine percent of the athletes suffered an injury in a previously injured area, reflecting the need for complete injury healing. The average injury found was 2.64 per 1000 h. Three contact sports presented the highest injury rate: soccer (7.21), judo (4.82), and basketball (4.31). The most common injuries were lumbar muscle strains (12.24%), ankle sprains (11.98%), and bone fractures (9.31%). By anatomical region, the three areas that concentrated the highest number of injuries were ankles (36.12%), knees (19.32%), and shoulders (6.47%). Fifty-nine point two eight of the injuries occurred during practices, and 40.72% during competition or peri-competition.

Furthermore, increased IR was associated (in that order) with: (a) Greater number of hours of practice per week. (b) Do not perform warm-ups. (c) Using inadequate sports facilities. (d) Being aged 14–17. (e) Do not perform physical preparation. (f) Inappropriate training load. (g) Do not perform injury-preventive activities. (h) Performing sports technique without the supervision of one sports coach. (i) Inadequate sports equipment. Since many of these factors are modifiable, preventive programs must be designed and implemented to reduce both the number of injuries and the injury rate. In contrast, sex, competitive level (amateur or professional), and following a healthy, balanced, and individualized diet did not influence IR. Moreover, being under stress could increase the IR, and being overweight the ID and ISS.

### Practical Applications

Since the number of injuries suffered by adolescent athletes and their injury rates still remains high, it is necessary to continue implementing prevention programs to make youth sports practice safe and healthy. Particular attention should be paid to prevent injuries in the following programs: injuries in contact sports, lower-body injuries, and injuries occurred during sports competitions. Preventive programs should also focus on modifiable injury risk factors, such as performing appropriate warm-ups, using adequate sports facilities, performing sports-specific physical preparation, adapting the training load to the athlete’s ability, performing injury-preventive activities, performing sports techniques under the supervision of a sports coach, and exercising with adequate sports equipment.

## Figures and Tables

**Figure 1 ijerph-18-04857-f001:**
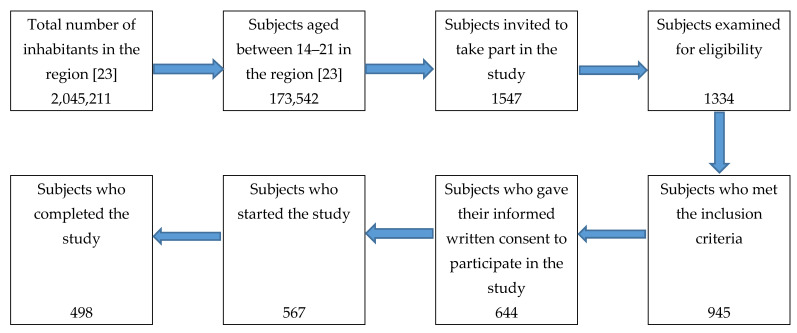
Flow Chart for selection of study participants.

**Table 1 ijerph-18-04857-t001:** Items included in the checklist used to collect the data.

Dimension	Item
Personal data	■Age ■ Sex ■Weight ■Height
Sports aspects	■Competitive level (amateur or professional) ■Sport practiced ■Weekly hours of practice ■Adequate sport-specific physical preparation ■Appropriate sports injury prevention programs ■Adequate warm-up ■Adequate sports equipment ■Adequate sports facilities ■Adapted training load ■ Sports technique supervision
Injury-related aspects	■Injury occurrence in 2019 ■Injury suffered ■Body part injured ■Injury recidivism or relapse ■ Injury duration ■ Injury severity ■ Event type (competition, peri-competition, training) ■Injury mechanism
Lifestyle	■Adequate nutrition and ■stress level

**Table 2 ijerph-18-04857-t002:** Sample characteristics and epidemiological aspects.

Sample characteristics	Sample size: *n* = 498 (Male: 343; Female 155)
	Age: 16.39(2.24); Male = 16.91(2.21); Female = 15.36(1.92)
	BMI: $\bar{x}$ = 22.13(4.71); Male: $\bar{x}$ = 22.92(5.31); Female: $\bar{x}$ = 20.30(1.96)
	Professional: *n* = 99; Amateur: *n* = 399
Sports practiced by the study participants	Soccer: 27.54%; swimming: 13.95%; weight training: 9.62%; athletics: 8.98%; basketball: 7.44%; tennis: 6.43%; judo: 4.75; paddle tennis: 4.36, volleyball: 3.95%; cycling: 3.11%; other: 9.87%
Injury rate (per 1000 playing and training hours)	$\bar{x}$ = 2.64 Soccer 7.21; Judo 4.82; basketball 4.31; volleyball 2.64; athletics 2.35; paddle tennis 1.72, tennis 1.39; weight training 1.12, cycling 0.59; swimming 0.35
Percentage of subjects who suffered an injury in 2019	40.4% (201 subjects): amateur: 38.8% (155 subjects); professional: 46.4% (46 subjects)
Percentage of subjects who suffered an injury in a previously injured area	39% (78 subjects): amateur: 37.4% (58 subjects); professional: 43.5% (20 subjects)
Most common injuries	Lumbar muscle strain 12.24%; ankle sprain: 11.98%; bone fracture: 9.31%; patellar tendonitis: 9.06%; abrasion (skin): 6.19%; muscle strain tear: 6.03%; anterior cruciate ligament tear: 4.72%; bone stress injury: 4.66%; internal lateral ligament tear: 4.59%; muscle contusion: 4.51%; Achilles’ tendonitis: 3.46%; bursitis 3.34%; cartilage injury: 3.22%; join sprain: 3.13%; tendinopathy: 3.03%; epicondylitis: 1.71%; posterior cruciate ligament tear: 1.51%; other: 7.25%
Body part injured	Ankle: 36.12%; knee: 19.32%; shoulder: 6.47%; foot: 5.39%; hand: 4.39%; lumbar-sacral-spine-buttock: 4.31%; thigh: 4.26%; wrist: 4.22%; thoracic-spine-upper back: 3.34%; elbow: 2.26%; forearm: 2.11%; hip-groin: 2.06%; lower leg: 1.03%; other: 4.42
Event type	Training: 59.28%; competition: 25.75%; peri-competition: 14.97%
Injury mechanism [21]	No identifiable single event: 30.64% Acute non-contact trauma: 21.54% Direct contact with an object: 20.48% Direct contact with another athlete: 12.01% Following contact with another athlete: 7.94% Following contact with an object: 7.39%

BMI: Body mass index; X bar, $\bar{x}$: Mean.

**Table 3 ijerph-18-04857-t003:** Injury duration, injury severity score, and injury rate as a function of different factors and conditions.

Factor		ID (Mean (SD))	ISS (Mean (SD))	IR (Mean (SD))
Sex	Male	4.71(8.66)	2.57(1.59)	2.81(3.37)
	Female	3.96(10.49)	2.56(1.41)	2.56(4.04)
Age (years)	14–17	4.49(10.07) *	2.66(1.53) *	2.86(3.67) *
	18–21	3.90(6.59)	2.52(1.55)	1.95(2.48)
BMI	UW	3.04(6.97)	2.55(1.57)	3.25(1.06)
	NW	2.72(2.93) *	2.21(1.41) *	2.34(0.24)
	OVW	5.04(10.06)	2.71(1.54)	3.06(0.79)
Competitive level	Professional	3.54(5.31)	2.65(1.45)	2.59(2.09)
	Amateur	4.72(9.97)	2.54(1.56)	2.94(2.63)
Weekly hours of practice	Less than 10 per week	3.16(4.47) *	2.47(1.55) *	2.41(2.64) *
	10 or more per week	4.83(10.04)	2.91(1.40)	3.56(2.91)
Performing an individualized and sport-specific physical preparation	Yes	3.52(6.65) *	1.92(1.14) *	1.74(2.13) *
	No	4.68(9.67)	2.69(1.56)	2.82(3.78)
Performing specific activities aiming to prevent injuries	Yes	3.19(5.26) *	2.25(1.29) *	2.31(2.58) *
	No	4.87(10.11)	2.65(1.59)	2.74(3.83)
Performing warm-ups adapted to athletes´ abilities and specific characteristics of the upcoming activity	Yes	2.58(3.15) *	2.21(1.03) *	2.59(3.69) *
	No	4.74(9.74)	2.61(1.58)	2.98(2.58)
Using sports equipment in good condition and frequently inspected	Yes	3.26(5.64) *	1.58(0.79) *	2.39(3.45) *
	No	4.62(9.54)	2.65(1.55)	2.66(3.59)
Using sports facilities in good condition, free of obstructions, well-lit, ventilated, and with adequate flooring	Yes	3.26(5.64) *	1.94(1.13) *	2.61(3.62) *
	No	4.65(9.57)	2.64(1.56)	2.89(3.19)
Training load adapted to athlete´s ability	Yes	4.15(8.18) *	2.15(1.76) *	1.76(2.61) *
	No	6.80(14.74)	2.63(1.50)	2.76(3.68)
Correct execution of sports techniques supervised by one sports coach	Yes	3.70(8.18) *	2.07(1.54) *	2.33(3.82) *
	No	4.88(9.72)	2.69(1.52)	2.71(3.53)
Following a healthy, balanced, and individualized diet	Yes	3.52(5.81)	2.47(1.51)	2.29(3.52)
	No	5.32(11.36)	2.69(1.56)	2.95(3.76)
Being subjected to stress due to the pressure received from family members, coaches, friends, or self-imposed	Yes	4.01(6.44)	2.53(1.52)	3.63(5.87)
	No	4.61(9.82)	2.57(1.54)	2.38(2.64)

ID: Injury duration (expressed in weeks). ISS: Injury severity score (rated from 1 to 5, with 1 being minor injury and 5 severe injury. IR: Injury rate (reported as the number of injuries per 1000 athlete-exposures: practice and competition). * Significant differences observed. BMI: Body mass index. UW: Underweight. NW: Normal weight. OVW: Overweight.

**Table 4 ijerph-18-04857-t004:** Correlation between independent variables and ID, ISS, and IR.

Factor	ID	ISS	IR
Age	r = −0.04; CI 95 % [−0.161, 0.153]; *p* = 0.61	r = −0.09; CI 95 % [−0.166, 0.148]; *p* = 0.24	r = −0.182; CI 95 % [−0.336, −0.028]; *p* = 0.031 *
BMI	r = 0.09; CI 95 % [−0.149, 168]; *p*= 0.22	r = 0.01; CI 95 % [−0.156, 158]; *p* = 0.82	r = 0.058; CI 95 % [−148, 160]; *p* = 0.467
Weekly hours of practice	r = 0.01; CI 95 % [−0.156, 0.158]; *p* = 0.86	r = 0.22; CI 95 % [−0.133, 0.177]; *p* < 0.001 *	r = 0.253; CI 95 % [0.096, 0.409]; *p* = 0.001 *
Carrying out an individualized and sport-specific physical preparation	r = −0.114; CI 95 % [−0.271, 0.043]; *p* =0.154	r = −0.20; CI 95 % [−0.174, 0.135]; *p* = 0.01 *	r = −0.277; CI 95 % [−433, −121]; *p* = 0.003 *
Performing specific activities aiming to prevent injuries (stretching, strength, proprioception, stabilization and muscular balance exercises)	r = −0.24; CI 95 % [−0.181, 0.133]; *p* < 0.001 *	r = −0.23; CI 95 % [−0.176, 0.130]; *p* = 0.01 *	r = −0.231; CI 95 % [−0.387, −0.075]; *p* < 0.001 *
Performing warm-ups adapted to athletes’ abilities and specific characteristics of the upcoming activity	r = −0.13; CI 95 % [−0.167, 0.141]; *p* = 0.11	r = −0.22; CI 95 % [−0.179, 0.135]; *p* < 0.001 *	r = -.217; CI 95 % [−0.374, −0.060]; *p* = 0.012 *
Using sports equipment in good condition and frequently inspected	r = −0.06, CI 95 % [−0.164, 0.151]; *p*= 0.45	r = −0.15; CI 95 % [−0.171, 0.141]; *p* = 0.05 *	r = −0.038; CI 95 % [−0.161, 0.155]; *p* = 0.638
Using sports facilities in good condition, free of obstructions, well-lit, ventilated and with adequate flooring	r = −0.03; CI 95 % [−0.161, 0.155]; *p* = 0.68	r = −0.07; CI 95 % [−0.164, 0.149]; *p* = 0.37	r = 0.03; CI 95 % [−0.155, 0.160]; *p* = 0.705
Training load (weekly frequency of practice, volume, density, intensity and recovery time) is adapted to athlete’s ability *	r = −0.15; CI 95 % [−0.170, 0.139]; *p* = 0.05 *	r = −0.13; CI 95 % [−0.169, 0.143]; *p* = 0.11	r = −0.282; CI 95 % [−0.439, −0.125]; *p* < 0.001 *
The correct execution of sports techniques is supervised by a sports coach	r = −0.27; CI 95 % [−0.184, 0.130]; *p* < 0.001 *	r = −0.24; CI 95 % [−0.181, 0.133]; *p* < 0.001 *	r = −0.263; CI 95 % [−0.421, −0.105]; *p* < 0.001 *
Following a healthy, balanced and individualized diet	r = 0.06; CI 95 % [−151, 0.163]; *p* = 0.45	r = −0.01; CI 95 % [−0.156, 0.155]; *p* = 0.88	r = 0.126; CI 95 % [−0.032, 0.283]; *p* = 0.114
Being subjected to stress due to the pressure received from family members, coaches, friends or self-imposed	r = −0.002; CI 95 % [−0.159, 0.158]; *p* = 0.98	r = −0.03; CI 95 % [−0.158, 0.152]; *p* = 0.68	r = 0.196; CI 95 % [0.42, 0.350]; *p* = 0.013 *

r: Pearson correlation; *p*: significance level was set at < 0.05; CI: Confidence Interval; *: significant correlation observed.

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
