# Peer review of "Epidemiology of Sports-Related Injuries and Associated Risk Factors in Adolescent Athletes: An Injury Surveillance"

_ijerph, 2021, doi:10.3390/ijerph18094857_

Round 1
Reviewer 1 Report
The article is well written and little wrong, and I list some small comments below that the authors can address to improve their manuscript.
Introduction section – can the authors develop their introduction section to strengthen the aim for this study. Why was this study conducted and what will conducting the study help develop in knowledge of the epidemiology of sport-related injuries?
Results section - a deeper analysis of the results is missing
Discussion section – can the authors develop some future directions that this research should examine.
Author Response
The authors would like to thank the editor and the reviewers for their helpful comments and suggestions and for their assistance in improving the quality of the manuscript.
Please see the attachment

Reviewer 2 Report
The authors did a great job. I believe after some major edits and modifications, this manuscript can be published. I have attached my comments and suggestions.
Abstract:
- It is recommended that a sentence that starts as a number, that the number is spelled out (example: line 17: Four hundred ninety-eight, instead of 498). This is done throughout the manuscript. The sentenced can be organized so that the authors can keep the numerical format.
- In the abstract, the average injury rate was stated as 2.64, however it is not clarified what the unit for injury rate is measured in. Perhaps the authors can state it as “2.64 per 1000 hours…”
- Authors give two list of injuries. The first being one indicating the “most common injuries” and the second being, “highest number of injuries”. Could the authors clarify the difference between the two lists? In the results it is stated as the most common injury type and the anatomical location of the injury, but it can be misleading as the “most common” does not the match the location. Are these separate evaluations?
- It may also be beneficial to include total amount of injuries reported so that the injury rate and percentages can be calculated if needed.
Introduction:
- Line 36, The authors can remove “Unfortunately,” and simply state “However,”. Additionally, the same sentence needs a citation as a claim is being made between sports practice and appearances of injuries.
- The introduction focuses on other counties and regions “USA and Africa”, but there is a no mention of Spain. Considering the population of the study is focused on Spain, it would be relatable to do so. If there is little known about the healthcare cost, the authors should make note of the this in the introduction.
- Lines 44-45, the authors cite a study that states, “….wherein the annual statewide cost estimated of high school…”, is “statewide” per each state in the country is that averaged across the United States.
- Lines 49-50: The statement is too wordy. The authors can simply state that there is a positive relationship between the total number of injuries and sports such as soccer.
- Lines 50-52: a citation is needed for the statement made.
- Lines 59-61. A potential reasoning should be made as to why adolescence have a clear tendency to abandon sports practice.
- Lines 62-64: the authors stated adolescents is divided into 3 phases, “early, middle, and late”. However, in the current study, the authors only surveyed middle and late. Why not survey all three phases since this is what was stated in the introduction?
- Throughout the study, the authors use gender. However, it is more appropriate to use sex, as there are more than two genders.
- Lines 74-76: the authors state “ Thus, more epidemiological studies that consider all these variables are required to improve scientific knowledge about sprots injuries and facilitate preventative interventions” , however in the current study only 2 groups were examined within 1 country. Therefore this is a limitation of this study.
- Lines 83-89: the authors list multiple variables as risk factors, but not provide a rationale for those variables.
Methods:
- Who is classified as a professional athlete? Is this someone that is getting paid?
- The authors used a check list item, however it is not stated that there the questionnaire is valid and reliable. Additionally, an appendix of the checklist should be provided. To assist with this, during the three online theoretical sessions that were held to unify criteria, the authors can collect the scores from the 3 sessions and run a reliability analysis. This assess both intra and inter-reliability.
- Did parents give consent for minors?
- Line 101-102, is this the most updated census? If so the participants make up about 25% of the population, is this correct?
- Paragraph 2.3- Were any of the athletes multisport athletes or where they only allowed to be single sport athletes?
- Paragraph 2.4 variables: There was a count of 23 not 24 variables. What is the difference between adequate sport-specific physical preparation and adequate warm up (items d&f from sports aspects 2). Once again the athletes were given a scale of 1-5, but there is no reported reliability or validation of the questionnaire, thus athletes may not be honest or answers may be inaccurate.
- Paragraph 2.5 Data sources and measurement: The authors report the use of Physical education and Sports teachers as research assistant. However, physical education instructors may not be qualified to judge performance or supervise practice or training. When assessing injuries, athletic trainers may have been more appropriate. Thus, this may have been a limitation to the study.
- Paragraph 2.6 Bias: It was stated that any injury that did not result in medical examination (148-149) was not considered in the injury rate analysis, however the scale is measured from 1-5 with 1 not requiring medical advice. Thus, it appears that there is some biased towards those who ranked an injury of 1. Was this the case or where all injuries used for analysis? Additionally, define medical advice. Who is giving this medical advice (physician, athletic trainer, school nurse etc.)?
- Paragraph 2.8 ethical clearance: Once again, was there minor consent given by parents?
Statistical analysis
- The authors do not indicate what software was used to run the statistical analysis ( JASP, SPSS, ETC.,)
- The authors do not mention what kind of ANOVA was used for analysis. Was it a 2 way mixed factorial ANOVA? Additionally, what did the models consist of and the levels for the variables?
Results:
- For clarification, is athletics the same as track and field?
- Could the authors give the total amount of injuries reported and then provide the injury rate per 1000 hours exposure.
- Lines 191-192: the authors list the top sports with the most reported injury, it would be beneficial to provide a percentage from the total injuries. (example: soccer (25%))
- Lines 194-195: the authors do not report if the differences between professional subjects versus armatures are significant.
- Line 195-196: The authors can remove “besides” and provide an exact percentage of the subjects that suffered an injury in a previous area. Was this significant (p < 0.05)?
- Throughout the results section, the authors use estimated values instead of exact values (example: line 200 , “ Almost 60%”, what is the exact value?)
- Did subjects have multiple injuries, aside from re-injuring themselves?
- Table 1.:
- Percentage of subjects who suffered an injured in a previous injured area: the authors do not provide a % after the values.
- Is peri-competition the same as competition?
- Out of the whole sample size, how many athletes were actually reported an injury?
- Per individual, how many injuries did they report?
- Lines 205-215: The authors do mention what ID stands for.
- Lines 205-215: BMI is not a great way to make comparisons between individuals as athletic populations are capable of having more musculature than body fat, therefore increasing BMI and putting them in the overweight category. BMI has been shown to be a poor indicator of performance compared to subcuatenous fat assessment ( Wallner-Liebmann et al., 2013, Krushitz et al., 2013)
- Line 211-213: authors do not indicate which group (normal vs overweight) had greater ISS
- Line 215: Remove “Besides,”. This is not a formal way of writing.
- Line 234-236: Were athletes who were supervised have a lower ID, ISS, IR than those who were not supervised?
- Line 239: C) what is considered a healthy, balanced, and individualized diet? Was there a food log that kept tract of the athlete’s diet or is this just an assumption?
- Table 2.
- The authors do not clarify the values as being reported as Mean (SD) or standard error of measurement (SEM).
- In the description and table, the following symbols are not needed as it is implied they are different from each other: # and Δ. This will also help use one symbol (*) to establish significant differences between groups for ID, ISS, IR. Additionally, instead of having three different symbols to establish differences for ID, ISS, IR, * can be placed next to values (mean (SD)), so that the audience knowns which groups and dependent variables are significantly different.
- Line 258-259: the authors state a positive correlation existed between ISS and weekly hours of practice, but do not provide the “r-value” nor the p value.
- Table 3.
- The authors do not include “IR” in the heading of the table.
- There is no symbol indicating significant correlations.
Discussion
- Throughout the discussion, the authors make comparisons between their results and other studies, however they fail to give interpretations or hypothesize about the results.
- Line 273-275: the authors could simply put the 48% in parentheses after 6 months, instead of having,” which raised to 48.8%”.
- Line 278: change “registered” to “reported”
- Line 286-289: the authors report previous findings, however when discussion the NCAA it is important to list out the division (DI, DII, or DIII).
- Line 292: the authors report that soccer had the highest rate of injury? Why do you think this may be the case? Could it be that soccer is the most popular sport in the area and, thus, there’s a potential skewness for injury.
- Line 299-300: Why do the authors think there may be a difference between injury rate for basketball?
- Line 301-302: For clarification, could the authors provide both the absolute and % for the most common injuries.
- Lines 317-319: Please elaborate upon this statement. Perhaps, the increase in injury rate could be due to poor sport programming, training, or years of activity wearing down the body.
- 2 Gender
- Provide more than one citation for lines 321-322, as the authors stated that it has been “frequently” found that males have a higher injury rate than females.
- Please clarify what socialization process means.
- The authors can remove “in one research carried out with” from line 325 as a way to reduce wordiness.
- 3 Age
- Please clarify why the authors think 14-17 year olds are having a greater ID, ISS, IR.
- 4 BMI
- Once again BMI may not be an appropriate way of assessing differences between athletes.
- Please clarify what is meant by “time loss injury” in line 340.
- The last sentence of the paragraph suggests that the greater absorption of forces leads to increases in injury. However, greater absorptions in forces do not necessarily lead to injuries but rather inability to dissipate those forces may lead to increases in injury risk. Greater body masses may put more shear forces on the bones and tendons if the individual is not strong enough to counteract the forces. With athletes, it may just be that individuals do not have proper mechanics and thus have an internal mechanical failure (muscles, tendons, ligaments) that cause injury.
- 5 Competitive level
- Line 351-352 states that it’s “that amateur athletes suffer fewer injuries” however, in the intro, younger athletes are more likely to be injured. If the statement stands, it could be that younger or amateur athletes have less hours playing professional athletes.
- 16 Limitations
- An additional limitation is the fact that this study is only conducted in one area so it may not generalizable to other regions.
- While the introduction discusses the 3 phases of adolescents, the authors only survey the middle and late phases.
- If the questionnaire is not reliable or validated it may not be an appropriate assessment tool.
- If injuries that reported “1’ were excluded, then there may be some biased towards the data.
Overall, this was a great study. With some revisions, I believe it will become publishable. There are some instances where there appears to be some language miscommunication, such as using transitional words/phrases. There is copious amount of data being reported and because of this, the tables appear to be too clustered and makes it difficult to interpret. The discussion does cover all the variables assessed; however, the authors do not interpret their results and provide rationales for their findings. I hope the comments and suggestions are helpful. I look forward to the next submission.
Author Response
The authors would like to thank the editor and the reviewers for their helpful comments and suggestions and for their assistance in improving the quality of the manuscript.
Please see the attachment.

Reviewer 3 Report
The study “Epidemiology of Sports-Related Injuries and Associated Risk Factors in Adolescent Athletes: An Injury Surveillance” is interesting, trying to identify patterns in injured adolescent athletes from a Spanish region. Nevertheless, the manuscript has some issues. The writing style and structure need to be revised since sometimes it seems that data is just copy-pasted, which is not adequate for a research article. Below, some additional issues and suggestions that need to be addressed:
- In the abstract, line 14, define the order for presentation of the factors influencing the higher injury rates.
- At least the second half of the abstract needs English language revision
- Some acronyms are not defined, e.g. BMI
- This topic is well addressed in the literature. Several works can be found, and the authors should compare the results from other studies with the patterns here found. Therefore, the authors should improve the introduction and use this expansion to improve the discussion.
- Line 88, one of the goals is to analyze the incidence of different risk factors on the reported injuries. It would be interesting for this study, especially to stand out from others in the literature, to answer the question on how these determined factors lead to the injury, which means going deeper into the injury mechanism. Some studies can help the authors in this aspect:
- DOI: 10.1177/0954411915592906
- DOI: 10.1136/bjsm.2005.018341
- DOI: 10.1177/1460408616676503
- DOI: 10.3109/02699052.2013.865269
- DOI: 10.1136/bjsm.2005.018341
- DOI: 10.2519/jospt.2008.2609
- Section 2, and its subsections reveal an inadequate structure. The subsections seem at some point mere bullet points. The structure of the methodology should be improved in terms of formatting.
- Subsection 2.4 is the perfect example of blocks of pasted text. The sentences need to be rewritten or at least presented in a different way, e.g. tables or bullet points.
- Line 157, there are some issues with the equation, and this should be indicated as an equation separated from the text. This comment is valid for all the other examples found within the text. Also, revise formatting of mathematical symbols (e.g. line 178, line 180, etc.)
- The last row of table 1 can be used by the authors to explore comment 5.
- There are several examples where values inferior to 1 do not have a 0 before the decimal separator. Please, revise the entire manuscript.
- Discussion and Conclusions should be improved based on previous comments.
Author Response

(The authors gave the same response as above.)

Reviewer 4 Report
The aim of this study was to determine the epidemiology of sport-related injuries in adolescent athletes aged 14 to 21 years as well as identify the risk factors associated with these injuries. The topic of the article is interesting. The article has been well designed and executed to answer the research questions. However, some questions could be discussed or clarified to improve the paper. In addition, the submission is not free from minor editorial errors.
ABSTRACT
The Abstract summarizes of what was done and what was found the paper in sufficient details.
INTRODUCTION
The Introduction provided information about the prevalence of sport-related injuries and their importance for the public and athletes’ general health. The authors well explained the scientific background and rationale for the investigation being reported. The goals of the study were precisely defined.
Some minor editorial remarks:
- Please add dot at the end of the sentence p. 2, line 56.
- Please define abbreviation BMI p. 2, line 85.
METHODOLOGY
This prospective injury surveillance comprised 498 athletes aged 14 to 21 years voluntarily participated in this study. Data was collected by 10 trained research assistants. The information collected included: personal data, sports aspects, characteristics of the injuries, and lifestyle. The statistical analyses used in the study are adequate.
Some questions, however, should be clarified or discussed:
- How many participants did not consent to study procedures or were excluded because did not meet other inclusion criteria?
- Please report numbers of participants at each stage of study - eg numbers potentially eligible, examined for eligibility, confirmed eligible, included in the study, completing follow-up, and analysed. This information could be included in the Methods or Results.
- Did you assess inter-rater reliability for collected variables?
RESULTS
The results has been shown in a clear way. However, please give information on potential confounders that may affect the results.
Some editorial remarks:
- Please add “=” (Table 1, Sample characteristics, Male (16.91)
- Please add “%” (Table 1, Percentage of subjects who suffered an injury in a previously injured area, 39.2)
- Please define abbreviation “Mean” as footnotes under the Table
- Please use “=” instead “:” in the phrase “ES: 0.103” p. 6, line 225.
- Please define abbreviation “BMI” as footnotes under the Table 2.
DISCUSSION
In the Discussion the Authors summarized key results with reference to study objectives. They also compared their results with other findings. The Discussion was divided into subsections which make this section clear and legible.
Minor remark: please correct the beginning of the sentence p. 10, lines 372-373.
The conclusions are supported by the results.
The literature is relevant and updated. However, according to the Instructions for Authors please use abbreviated journal name in the reference list.
Author Response
Dear reviewer 4. The authors would like to thank the editor and the reviewers for their helpful comments and suggestions and for their assistance in improving the quality of the manuscript.
Please see the attachment.

Reviewer 5 Report
The authors present the results of a survey made in a region of Spain about sports injuries in a sample of young people (14-21 years old). I have missed a scientifically elaborated discussion. The practical applications described by the authors are far from the results obtained. I also have other comments, which I divided into major and minor. I would encourage the authors to improve the writing of all sections, improving the presentation of the results, including better identifications of relations among variables, and rewriting the discussion and conclusions sections.
Major
- L28-30. These conclusions seem obvious, quite vague. What is the contribution of your paper?
- The only novelty of the paper seems to be the fact that the analysis was in Spain. However, this was in a region of Spain. I wonder if the results could be extrapolated.
- L167-168. Normally, the ethical committee is from the institution where the measurements or the intervention is performed. It is quite weird that the Ethical Committee is from another country. I am not sure if this is possible. Why not from Spain if the survey was done there?
- L194-195. How would you explain that? Non-professional athletes should suffer fewer injuries since their training is better, right?
- There are many results in the results section. However, there is no cross-correlation analysis, in the sense that probably people who have an individualized preparation also have better sports equipment? Or competitive level also correlates with the weekly hours of practice? I think that the relevant information could be summarized, and probably not all data on the left column of table 2 is relevant. All points of the discussion are quite repetitive. The discussion could be improved significantly.
- The authors mention that there are significant negative or positive correlations. However, all correlation values r are very low (in absolute value). This means that the correlation, despite being significantly positive or negative, is weak.
- The authors mention that one of the main findings was to verify that 40.4%. Was this an initial hypothesis to be verified? Why?
- Section 5.5. In lines 194-195 the authors mention that professional athletes suffered more injuries than amateurs, but in this section, they mention that there were no significant differences. Is this because the differences were not significant? I would write it clearer.
- Why amateur athletes should suffer fewer injuries if, a priori, their training is “poorer”?
- I think that a brief description or what is considered by professional athletes should be included.
- R values for ID, ISS, and IR against training load were negative. Would this mean that the higher the training load, the lower are the coefficients, right? But the authors in lines 412-414 mention that: “Subjects who used a significantly lower training load adapted to athlete's ability presented a significantly lower ID, ISS, and IR than those who did not.”. I think that I miss something.
- The first two paragraphs of the Conclusion are a repetition of the abstract.
- The practical applications of the Conclusions are quite vague, these are not in line with the results.
Minor
- In scientific papers, normally a sentence is not started by a numeric value. There are multiple cases throughout the text.
- The units of the injury rate are not clear. For instance, in line 20 page 1, what are the units of 2.64?
- There are many short paragraphs in the introduction. In a scientific paper, one paragraph should transmit an idea. It would be better for the reader.
- “The data”, instead of “To data”?
- Section 2.1. It seems quite weird a section with only one line.
- What was the population size N?
- “limp” should be “limb”.
- Table 2, what is the meaning of (weeks) between the parenthesis?
- Table 2. Meaning of UW, NW, OVW should be included in the table caption.
- Section 5.2. Since participants were humans, I think that it is more appropriate to mention men and women than males and females.
Author Response

(The authors gave the same response as above.)

Round 2
Reviewer 2 Report
The authors did a great job with their edits addressing my comments and concerns. I have no further comments.
Author Response
Dear Reviewer 2. The authors would like to thank the editor and the reviewers for their helpful comments and suggestions and for their assistance in improving the quality of the manuscript. Hereunder we detail the modifications made in the article. Please see the attachment

Reviewer 3 Report
Thanks for addressing most of my comments. Nevertheless, some issues remain:
- The structure of the methodology section continues to be odd. Now, it seems some information is not presented. Page 3 is blank (maybe some compiling issues), please provide the information missing.
- Equations need to be formatted accordingly. Please use variables, math symbols, etc instead of writing text in it. Then, describe the parameters in the text.
- The discussion was well improved. Unfortunately, the introduction needs additional work as indicated in the previous review.
Author Response
Dear Reviewer 3. The authors would like to thank the editor and the reviewers for their helpful comments and suggestions and for their assistance in improving the quality of the manuscript. Hereunder we detail the modifications made in the article. Please see the attachment
